# Research on High-Quality Development Evaluation and Regulation Model: A Case Study of the Yellow River Water Supply Area in Henan Province

**Xiuyu Zhang [1], Ying Zhou [1] and Chunhui Han [1,2,*]**

1   College of Water Resources, North China University of Water Resources and Electric Power, Zhengzhou 450046, China
2   Henan Key Laboratory of Water Resources Conservation and Intensive Utilization in the Yellow River Basin, Zhengzhou 450046, China
*   Correspondence: hanchunhui@ncwu.edu.cn; Tel.: +86-158-0381-0121

**Abstract:** Unbalanced and inadequate development has led to challenges such as resource shortage, ecological fragility, and economic backwardness in the water-receiving area of Henan Province, which has hindered the high-quality development of the region. This research aims to construct a high-quality development regulation model. The single-indicator quantification–multi-indicator synthesis–multicriteria integration (SMI-P) method was used to evaluate the level of regional development; using the method of embedded system dynamics, the regulation model was constructed and applied. The results show that (1) from 2011 to 2020, the degree of regional high-quality development showed an increasing trend, rising from a low level (0.34) to a medium level (0.57), an increase of one level, but the level of high-quality development still has a lot of room for improvement and faces great challenges; (2) the development of a resource–ecology–economy–social system was in a state of non-harmonious balance, and the proportion of contributions of each system was very different, but the system was developing towards a state of harmony and balance; (3) based on the constructed high-quality development regulation model, the key constraints affecting the high-quality development of the Yellow River water supply area in Henan Province are per capita water consumption, per capita energy consumption, carbon emissions per capita, GDP per capita, etc. The model constructed in this paper can improve the quality development level through the application of examples. The high-quality development degree of the study area was increased from 0.57 to 0.8.

**Keywords:** high-quality development evaluation; key constraints; regulation model; the Yellow River water supply area in Henan Province

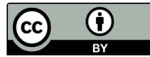

## 1. Introduction

The Yellow River Basin has been China's political, economic, and cultural center, and its ancient development level, represented by the Yellow River Basin, has long been ahead of the rest of the world. However, the flooding of the Yellow River is also severe [1], and generation after generation, Chinese have worked hard to explore and unremittingly strive for the Yellow River to benefit the Chinese nation forever. The governance of the Yellow River has new requirements in the new era. The new idea is to strengthen the construction of ecological civilization and take the road of high-quality development. On 18 September 2019, General Secretary Xi Jinping put forward the strategy of "ecological protection and high-quality development of the Yellow River Basin" (hereinafter referred to as the "Yellow River Strategy") in Zhengzhou and delivered an important speech [2], pointing out that the Yellow River Basin is an important ecological barrier, an important economic zone, and an important area for winning the battle against poverty. The Outline of Planning for Ecological Protection and High-Quality Development of the Yellow River

Basin [3] pointed out that the biggest contradiction in the current Yellow River Basin is the shortage of water resources, the biggest problem is ecological fragility, the biggest shortcoming is insufficient high-quality development, and the biggest weakness is the insufficient development of people's livelihood. It all boils down to the problem of unbalanced and inadequate development of the resource–ecological–economic-social system, so this paper examines it from a systems perspective.

At first, research on high-quality development focused on the economic field [4–7]. After the Yellow River Strategy was elevated to the fifth national development strategy, it attracted widespread attention from all walks of life. The academic community set off an upsurge in research on the high-quality development of the Yellow River Basin. Xia Jun [8] summarized the opportunities and challenges of comprehensive management and high-quality development of the Yellow River Basin; Zhang Gongsheng [9], An Shuwei [10], and Shi Bihua [11] all studied the connotation and path of ecological protection and high-quality development of the Yellow River Basin. Liu Changming [12] and Yang Yongchun [13] summarized the research priorities of high-quality development. With the further application and research of the concept of high-quality development, the concept of high-quality development in more fields has been proposed and studied, such as agriculture [14], land [15], high-quality water conservancy development [16], etc. With the deepening of research, Zuo Qiting [17] proposed a research framework for strategy implementation based on the needs of strategy implementation. Xu Yong [18] built the overall framework of the national strategy for ecological protection and high-quality development of the Yellow River Basin according to the three logical progressive links of "foundation-ecological priority, bearing-development constraint, and drive-internal and external linkage". Zuo Qiting [19] constructed a research framework for the optimization of high-quality development paths, analyzed the four key problems that need to be solved urgently and the three core technologies that need to be broken through, discussed the key contents of the path optimization research in detail from four aspects, and preliminarily conceived the corresponding research ideas.

There have been many quantitative studies on high-quality development, and different scholars have constructed evaluation index systems from different angles for high-quality development evaluation, constructed quantitative evaluation index systems [20,21], and evaluated the high-quality development level of the Yellow River Basin. At first, quantitative methods for high-quality development were relatively simple, mostly using linear weighting [22,23]. Zuo Qiting [24] put forward the "Five Criteria" for judging high-quality development, built a high-quality development evaluation system based on the judgment criteria, and evaluated the high-quality development level of the Yellow River Basin from 2008 to 2018. Zhang Jinliang [25] proposed a comprehensive assessment theory of basin development quality based on the Basin Development Index (BDI) and comprehensively evaluated the evolution state and development quality of river basin giant systems. However, there are relatively few studies from the perspective of the system, especially the complex system composed of resources–ecology–economy–society, and research from the perspective of the system can explore the interrelationship between the system as a whole and the elements that make up the system and can essentially explain the structure, function, behavior, and dynamics of the system and grasp the overall system to achieve the optimal goal.

There are relatively few studies on the regulatory model of high-quality development, and most of the current research on the regulatory model is aimed at the impact of a changing environment on the water resources system [26], water resources management [27], water resources allocation [28], harmonious regulation of people and water [29], etc., although Zuo et al. [24] constructs a high-quality development level evaluation model from the system perspective but only evaluates the historical level, and the key influencing factors affecting the high-quality development level are not clear. There is still a lack of regulatory model research on how to improve and enhance the level of high-quality development.

With the major implementation of the Yellow River Strategy and the increasingly serious problems in resources and ecology in the Yellow River Basin, determining how to put the Yellow River Basin on the road of high-quality development has become a common issue. This paper studies the complex system composed of resources–ecology–economy and society and first constructs an evaluation index system from the perspective of the system to evaluate high-quality development and explores the current development status of the region. Then, the regulatory model is constructed, and the key constraints affecting high-quality development are identified through the constructed regulatory model. Finally, by regulating key constraints and realizing the regulation of high-quality development level, this paper can promote the improvement of regional high-quality development level.

## 2. Materials and Methods

The research content flow of this paper is shown in Figure 1; this paper aims to construct a high-quality development regulation model to improve the level of regional high-quality development. The research content flow chart mainly contains two parts: the main research content and the methods used, which can clearly show the main ideas of this research.

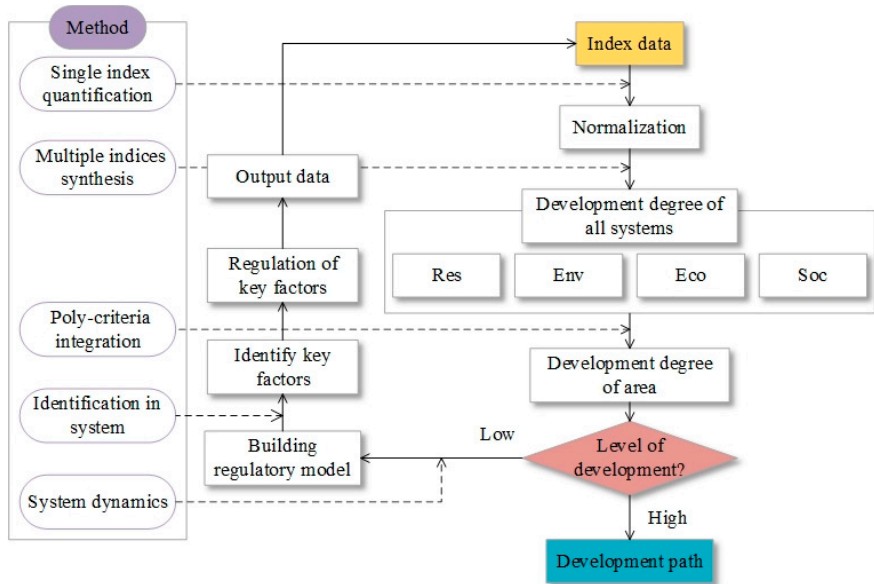

**Figure 1.** Study content flowcharts.

The main research process consists of two phases:

The first stage is the evaluation of the status quo of high-quality development based on the reasonable selection of evaluation indicators and suitable formulation of evaluation standards. The indicator data are collected, the indicators are normalized through single indicator quantification, and then the high-quality development degree of each system is calculated through multi-index comprehensively. Then, each region's final high-quality development degree is calculated through multicriteria integration, at which time the status quo evaluation of high-quality development is completed.

The second stage is the construction and application of the high-quality development control model. When the degree of high-quality regional development is high, there is no need for regulation, and the current high-quality development path can be maintained; when the degree of high-quality regional development is low, it is necessary to construct a regulatory model to improve the high-quality regional development level. On the basis of clarifying the internal mechanism of each subsystem and composite system, the regulatory model is constructed through the method of system dynamics, and the key

constraints affecting high-quality development are identified by using the regulatory model. The relevant parameters of the key constraints of the model are used to obtain the index values after regulation under each scheme, which are re-replaced in the evaluation process of the first stage for evaluation, and the scheme with the highest degree of high-quality development is selected, and finally, the path that can improve the high-quality development degree of the region is optimized.

*2.1. Current Situation Assessment Methodology*

This paper comprehensively considers the regional characteristics of the Huanghuang water-receiving area in Henan Province and the urgent problems currently faced and adheres to the principles of scientificity and conciseness, the principle of combining completeness and representativeness, the principle of combining qualitative and quantitative analysis, and the principles of accessibility and operability [22]. Evaluation indicators are selected from the four subsystems of resources, ecology, economy and society; the positive index attribute is "+" and the negative index attribute is "−".The evaluation index system is shown in Table 1.

**Table 1.** High-quality development evaluation index system.

| Target Layer | System Layer | Metrics Layer | Unit | Attribute | Numbering |
|---|---|---|---|---|---|
| High-quality development degree | Resource system (Res) | Water production modulus | $10^4\,m^3/km^2$ | + | A1 |
| | | Precipitation | mm | + | A2 |
| | | Per capita water consumption | $m^3$/capita | − | A3 |
| | | Per capita energy consumption | t standard coal/capita | − | A4 |
| | | Per capita electricity consumption | KW·h/capita | − | A5 |
| | | Cultivated land per capita | $hm^2$/capita | + | A6 |
| | | Grain output per sown area | $kg/hm^2$ | + | A7 |
| | Ecology system (Env) | Wastewater treatment rate | % | + | B1 |
| | | Wastewater emissions per 10,000 yuan of GDP | $10^4\,m^3/10^4$ yuan | − | B2 |
| | | COD emissions per capita | t/capita | − | B3 |
| | | Consumption wastes harmless treatment rate | % | + | B4 |
| | | Green coverage rate of built-up district | % | + | B5 |
| | | Exhaust gas emissions per 10,000 yuan of industrial Added value | $t/10^4$ yuan | − | B6 |
| | | Carbon emissions per capita | t | − | B7 |
| | | NDVI index | / | + | B8 |
| | Economic system (Eco) | GDP per capita | yuan/capita | + | C1 |
| | | Annual growth rate of GDP | % | + | C2 |
| | | Proportion of tertiary industry | % | + | C3 |
| | | Unemployment rate | % | − | C4 |
| | | Water consumption per 10,000 yuan of industrial added value | $m^3/10^4$ yuan | − | C5 |
| | | Energy consumption per 10,000 yuan GDP | t standard coal/$10^4$ yuan | − | C6 |
| | | Number of patent applications granted | piece | + | C7 |
| | Social Systems (Soc) | Urbanization rate | % | + | D1 |
| | | Engel coefficient | / | − | D2 |
| | | The ratio of income of urban and rural residents | / | − | D3 |
| | | Density of expressways | $km/km^2$ | + | D4 |
| | | Years of schooling per capita | years/capita | + | D5 |

| Number of medical and health personnel per 10,000 Persons | capita/10⁴ capita | + | D6 |
| Popularizing rate of tap water | % | + | D7 |
| Gas coverage rate | % | + | D8 |

The method of "the single index quantization-multi index synthesis-multicriteria integration method" [29] (SMI-P) was adopted to evaluate high-quality development. The main calculation steps are as follows:

(1) The single index quantization. The fuzzy membership function is used to quantify a single index of different dimensions into values between [0, 1]. The positive and negative indexes are calculated according to the left and right sides of Formula (1), respectively.

$$\mu_{k1}=\begin{cases} 0 & , \ x_k \leq a_k \\ 0.3\left(\frac{x_k-a_k}{b_k-a_k}\right) & , \ a_k < x_k \leq b_k \\ 0.3+0.3\left(\frac{x_k-b_k}{c_k-b_k}\right), & b_k < x_k \leq c_k \\ 0.6+0.2\left(\frac{x_k-c_k}{d_k-c_k}\right), & c_k < x_k \leq d_k \\ 0.8+0.2\left(\frac{x_k-d_k}{e_k-d_k}\right), & d_k < x_k \leq e_k \\ 1 & , \ e_k < x_k \end{cases} \qquad \mu_{k2}=\begin{cases} 1 & , \ x_k \leq e_k \\ 0.8+0.2\left(\frac{d_k-x_k}{d_k-e_k}\right), & e_k < x_k \leq d_k \\ 0.6+0.2\left(\frac{c_k-x_k}{c_k-d_k}\right), & d_k < x_k \leq c_k \\ 0.3+0.3\left(\frac{b_k-x_k}{b_k-c_k}\right), & c_k < x_k \leq b_k \\ 0.3\left(\frac{a_k-x_k}{a_k-b_k}\right) & , \ b_k < x_k \leq a_k \\ 0 & , \ a_k < x_k \end{cases} \qquad (1)$$

where $\mu_k$ is the membership of indicator; $x_k$ is the indicator value; $a_k, b_k, c_k, d_k, e_k$ are the characteristic values of the indicator; $a$ is the worst value, expressed by the worst level of the index Yellow River Basin; $b$ is the difference value, which is determined by interpolation; $c$ is the passing value, expressed according to the average level of the Yellow River Basin and Henan Province; $d$ is the better value, which is determined by interpolation; $e$ is the best value, expressed by the optimal level of the index Yellow River Basin. The selection of the characteristic values of each indicator is shown in Table 2.

**Table 2.** The characteristic value of the indicator.

| Index | a | b | c | d | e | Index | a | b | c | d | e |
|-------|---|---|---|---|---|-------|---|---|---|---|---|
| A1 | 20 | 15 | 10 | 5 | 0 | C1 | 100,000 | 90,000 | 80,000 | 60,000 | 40,000 |
| A2 | 600 | 500 | 400 | 300 | 200 | C2 | 10 | 7.5 | 5 | 2.5 | 0 |
| A3 | 200 | 250 | 300 | 350 | 400 | C3 | 0.5 | 0.45 | 0.4 | 0.35 | 0.3 |
| A4 | 1 | 2 | 3 | 4 | 5 | C4 | 2 | 2.5 | 3 | 3.5 | 4 |
| A5 | 1000 | 2000 | 3000 | 4000 | 5000 | C5 | 15 | 20 | 25 | 30 | 35 |
| A6 | 0.087 | 0.067 | 0.047 | 0.030 | 0.013 | C6 | 0.2 | 0.4 | 0.6 | 0.8 | 1 |
| A7 | 6000 | 5250 | 4500 | 3750 | 3000 | C7 | 70,000 | 52,500 | 35,000 | 17,500 | 0 |
| B1 | 100 | 95 | 90 | 85 | 80 | D1 | 80 | 75 | 70 | 65 | 60 |
| B2 | 1 | 2 | 3 | 4 | 5 | D2 | 0.2 | 0.25 | 0.3 | 0.35 | 0.4 |
| B3 | 0.001 | 0.002 | 0.003 | 0.004 | 0.005 | D3 | 1 | 1.5 | 2 | 2.5 | 3 |
| B4 | 100 | 97.5 | 95 | 92.5 | 90 | D4 | 0.2 | 0.15 | 0.1 | 0.05 | 0 |
| B5 | 50 | 45 | 40 | 35 | 30 | D5 | 13 | 11.5 | 10 | 8.5 | 7 |
| B6 | 0 | 5000 | 10,000 | 15,000 | 20,000 | D6 | 200 | 150 | 100 | 75 | 50 |
| B7 | 0.02 | 0.04 | 0.06 | 0.08 | 0.1 | D7 | 100 | 97.5 | 95 | 92.5 | 90 |
| B8 | 1 | 0.9 | 0.8 | 0.7 | 0.6 | D8 | 100 | 95 | 90 | 85 | 80 |

(2) Multi-index synthesis. In order to make up for the shortcomings caused by single subjective empowerment or single objective empowerment, the entropy weight method and analytic hierarchy method are used to calculate the combined weight of each index. Then, multi-index weighting is carried out, and the formula is as follows:

$$G_t = \sum_{k=1}^{n} \omega_k \mu_k \tag{2}$$

where $G_t$ is the exponent of the system layer; $n$ is the number of indicators in each system layer; and $\omega_k$ is the indicator's weight relative to the system layer.

(3)　Multicriteria integration. The system layers are gathered together to calculate the final degree of high-quality development, and the formula is as follows:

$$F = \sum_{t=1}^{m} \omega_t G_t \tag{3}$$

where $m$ is the number of system layers; $\omega_t$ is the weight of the system layer.

(4)　Grading. The hierarchical division method in Reference [24] divides the degree of high-quality development into five levels and the level of development into five levels according to the degree of development: (0, 0.2) is lower, [0.2, 0.4) is low, [0.4, 0.6) is medium, [0.6, 0.8) is high, and [0.8, 1) is higher.

*2.2. Governance Model Construction*

　　This paper adopts the embedded system dynamics method to construct the control model of embedded system dynamics (ESD) proposed by Professor Zuo Qiting in 2007, which is based on the original system dynamics model, combined with the characteristics of its system; the professional module equations in its system are embedded into the original system dynamics model to form a new coupling model. In addition to retaining the advantages of the original system dynamics model, this coupling model also considers the problems of other research fields, which not only improves the application level of system dynamics but also provides an excellent solution to complex and highly specialized system simulation problems.

　　The ESD model was established using Vensim software. The professional module equations such as the water quantity calculation equation, wastewater discharge calculation equation, and total pollutant discharge calculation equation were embedded in the resource–ecology–economy–society system model (some formulas involved in the model are shown in Table 3). After embedding the variable equation into the model, the governance model can be obtained (Figure 2). The high-quality development degree of the yellow water-receiving area of Henan Province was predicted from 2016 to 2020 for model testing. After the model passes the test, the region's current situation and the resource, ecological, economic, and social development trend of the planning year are dynamically analyzed, and the high-quality development degree is compared and analyzed by setting different schemes. Finally, the optimal plan suitable for the high-quality development of the yellow water-receiving area of Henan Province is obtained.

**Table 3.** Variable equations in the governing model.

| Variable Equations | Unit |
| --- | --- |
| Total water consumption = agricultural water consumption + industrial water consumption + domestic water consumption + ecological water consumption | 100 million m³ |
| Agricultural water consumption = farmland irrigation water quota × cultivated land area | 100 million m³ |
| Industrial water consumption = industrial water quota × industrial added value | 100 million m³ |
| COD discharge = COD concentration × sewage discharge | t |
| Proportion of tertiary industry = tertiary industry GDP / GDP | % |
| Population change = total population × (birth rate－death rate) | person |

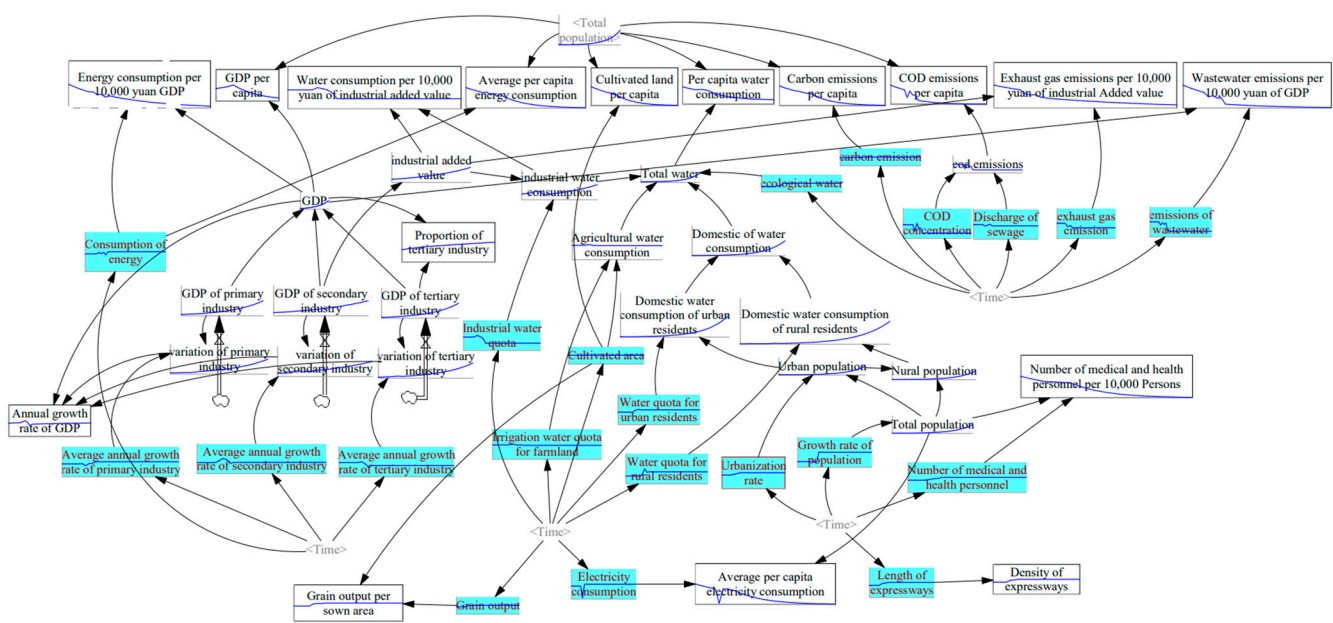

**Figure 2.** Governance model diagram.

After constructing the control model, the model test needs to be carried out, and the evaluation index values from 2016 to 2020 are derived for model testing. The year 2015 is selected as the base year, the simulation time step is one year, and the end time is 2020. If the error of the simulation results is within 5%, the model passes the validity test. This paper selects the simulation values of the three main variables of total water consumption, GDP, and total population for validity testing (Table 4). It can be seen from the table that the simulation results of the three main variables are within 5%, which can meet the calculation needs, and the model passes the test.

**Table 4.** The test results of the model.

| | GDP (10⁸ Yuan, 10⁸ Yuan, %) | | | Water Use (10⁸ m³, 108 m³, %) | | | Total Population (10⁴ People, 10⁴ Persons, %) | | |
|---|---|---|---|---|---|---|---|---|---|
| Yea r | Simulation Value | Actual Value | Relative Error | Simulation Value | Actual Value | Relative Error | Simulation Value | Actual Value | Relative Error |
| 2015 | 29,706.5 | 29,706.45 | 0 | 169.338 | 169.518 | −0.11 | 6880 | 6880 | 0 |
| 2016 | 32,523.1 | 32,523.06 | 0 | 172.343 | 172.745 | −0.23 | 6918.87 | 6918 | 0.01 |
| 2017 | 35,673.9 | 36,002.71 | −0.91 | 177.721 | 178.859 | −0.64 | 6961.42 | 6944 | 0.25 |
| 2018 | 39,572.6 | 38,806.04 | 1.98 | 177.545 | 176.717 | 0.47 | 7003.05 | 6987 | 0.23 |
| 2019 | 42,720.3 | 43,476.26 | 1.74 | 182.106 | 179.264 | 1.59 | 7037.51 | 7017 | 0.29 |
| 2020 | 44,081.9 | 43,824.24 | 0.59 | 173.665 | 174.426 | −0.44 | 7066.92 | 7409 | −4.61 |

### 2.3. Study Area and Data

After implementing the Yellow River strategy, the Henan provincial government proposed to "take the lead in setting Henan benchmarks in the whole river basin". It is urgent to explore a new way of high-quality development with regional characteristics. In this paper, the area in Henan Province that refers to the Yellow River water is taken as the research object, that is, the receiving area of the Yellow River in Henan Province (Figure 3). Located on both banks of the Yellow River in the north of Henan Province, between 33°85′–36°10′ N latitude and 110°21′–116°39′ E longitude, Henan Province is the core area of China's Silk Road Economic Belt and Belt and Road Economic Belt. Although the geographical location of Henan Province is superior, there are also a variety of problems in

the region, and the problem of unbalanced development of the regional resources–ecology–economy–social system is prominent.

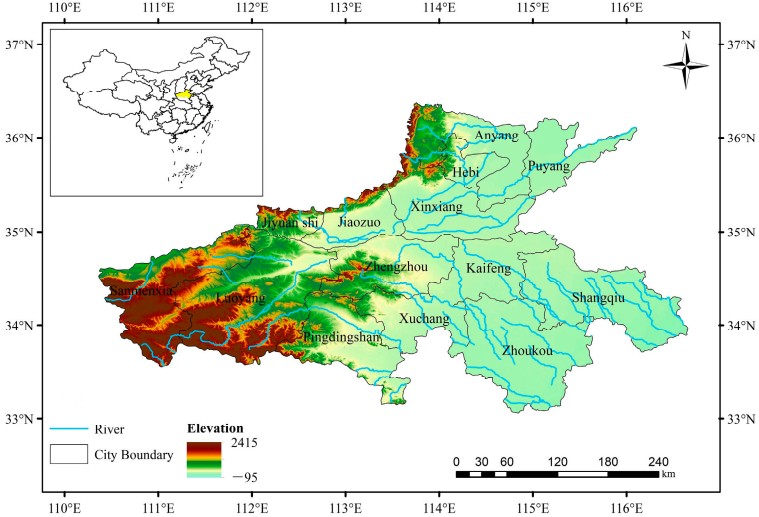

**Figure 3.** Schematic diagram of the location of the study area.

The research data are mainly derived from the 2011–2020 Henan Statistical Yearbook, Henan Provincial Water Resources Bulletin, Henan Provincial Environmental Statistics Annual Report, and water resources bulletins of various cities in the Huanghuang water-receiving area of Henan Province.

## 3. Results and Discussion

### 3.1. Status Quo of High-Quality Development

According to the above, that the high-quality development degree of the yellow water-receiving area of Henan Province can be calculated from 2011 to 2020 (Figure 4). The overall high-quality development degree of the yellow water-receiving area of Henan Province fluctuates and rises, and the overall level rises from a low (0.34) to a medium level (0.57), an increase of one notch.

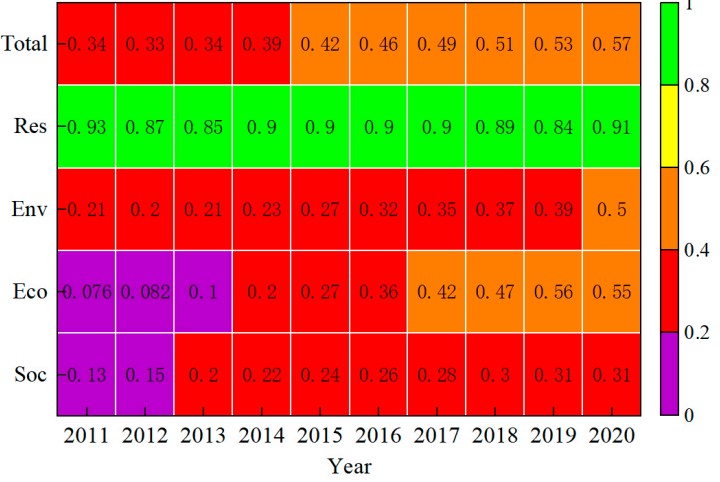

**Figure 4.** Changes in the degree of high-quality development of regional systems and the whole.

The high-quality development of regional resource subsystems is relatively stable and has always been at a high level. In 2019, the degree of high-quality development of resource subsystems decreased significantly. The analysis data showed that the decrease

in cultivated land area and grain output in 2019 led to a decrease in the high-quality development of the resource system. After the "Yellow River Strategy" was proposed in 2019, the per capita cultivated land area increased. The per capita water consumption decreased, and the high-quality development of resource subsystems in various local-level cities in 2020 was significantly improved. The degree of high-quality development of the resource system has always fluctuated and changed more than the overall high-quality development degree of the region, indicating that the resource system of the Huanghuang receiving area of Henan Province attaches more importance to the conservation, intensive use, and sustainable use of resources. However, there is a downward trend, and seeking a stable development model is still necessary.

The degree of high-quality development of regional ecological subsystems showed a slow upward trend. Following the official implementation of the newly revised Environmental Protection Law of the People's Republic of China in 2015, the region has strengthened ecological environment protection measures, the sewage treatment rate increased significantly, the per capita COD emission was significantly reduced, and the high-quality development degree of regional ecological subsystems was significantly improved in 2016. After the "Yellow River Strategy" was proposed in 2019 and China put forward the "dual carbon goals" in 2020, various cities further strengthened their ecological and environmental protection measures, significantly reduced wastewater discharge and carbon emissions, improved regional vegetation coverage, and greatly improved the high-quality development of ecological subsystems in various cities in 2020. In the ten years from 2011 to 2020, the high-quality development of regional ecological subsystems was greatly improved, and the overall increase was 1~2 levels. The overall high-quality development degree of the region is higher than that of the ecosystem. However, the high-quality development of the ecosystem improved faster, indicating that the Huanghuang water-receiving area of Henan Province attaches great importance to ecological issues. Attention has been paid to water quality, air quality, and urban greening. However, the high-quality development degree of ecosystems in 2020 is still only at a medium level, and ecological problems need continuous attention.

The degree of high-quality development of regional economic subsystems is rising, and the increase is significant. The development of regional green industries is good; the water consumption of 10,000 yuan of industrial added value and energy consumption of 10,000 yuan of GDP has decreased significantly, so the degree of high-quality development of economic subsystems has greatly improved. However, the current industrial structure is a single structure, and the innovation ability is insufficient. There is still much room for improvement. The overall high-quality development degree of the region is always better than the high-quality development degree of the economic system. There is a big gap between the advanced regions in the country, which needs further improvement.

The degree of high-quality development of regional social subsystems shows a slow upward trend, but the increase is not significant. From 2011 to 2020, the regional tap water penetration and gas penetration rate increased significantly. Now, they have reached the whole popularization, so the high-quality development of regional social subsystems is slowly rising. However, the income ratio of urban and rural residents, the number of medical and health personnel, and the per capita years of education have not improved greatly, so the high-quality development of social subsystems is restricted, and the improvement is not considerable. The overall high-quality development degree of the region is significantly higher than the high-quality development degree of the social system. The high-quality development degree of the social system is always at a low level or below, indicating that education, medical care, employment conditions, etc. need to be continuously improved. The social security system needs to be continuously improved to meet people's needs for a better life.

The contribution ratio of each system to the high-quality regional development of that year is shown in Figure 5.

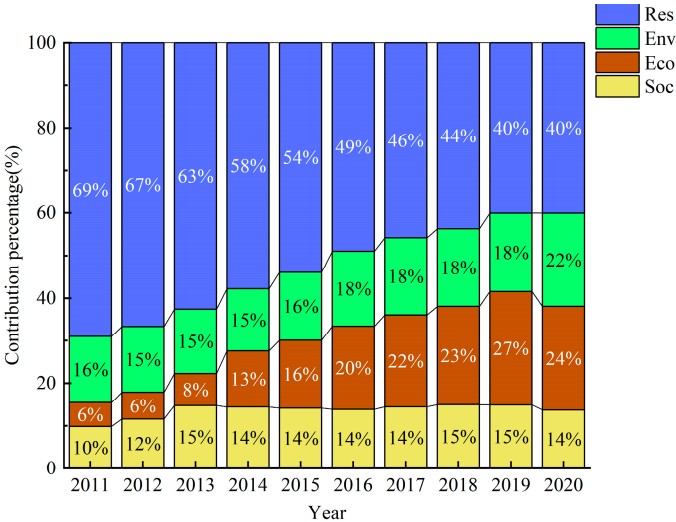

**Figure 5.** The contribution ratio of each subsystem to high-quality region development.

When the development of the system is in a harmonious and balanced state, each subsystem should contribute 25%. As seen in Figure 5, the current development of the system is in a non-harmonious equilibrium state, and each system's contribution is very different. However, the resource–ecological–economic–social system is developing towards a harmonious and balanced state. The degree of high-quality development of resource systems remains unchanged, and the proportion decreases, indicating that the level of other systems has improved; the proportion of ecological and economic systems has increased, indicating that its high-quality development degree has improved. The proportion of social systems remains unchanged, indicating that their high-quality development degree is slow to improve.

### 3.2. Governance Model Application

### 3.2.1. Identification of Key Constraints

Many phenomena in life are not independent but interact with and influence each other. The emergence of a result is often the result of multiple factors and links. If the influence of only one of them is examined in light of other factors, the conclusions may become one-sided or even wrong. In the high-quality development regulation model, the indicators in the system have a complex mutual influence relationship. The change in one indicator may cause changes in other indicators. In order to more accurately identify the parameters that have more significant impact on the index value in the whole system, the critical constraints of high-quality development are identified using the method of regulation and identification in the system. The main idea is to only regulate the value of a particular parameter at a time, record the influence of the parameter on the value of other indicators, output all index values in the system after regulation, and re-evaluate high-quality development. If the parameter change significantly impacts the evaluation results, the indicators related to this parameter are the key constraints affecting high-quality development.

There are 20 parameters related to the evaluation index in the above regulation model. When the system is regulated and identified, the relevant parameters of the positive index are expanded to 1.25 times the original, and the relevant parameters of the negative index are reduced by 0.75 times. The results after re-evaluation of high-quality development are recorded in Table 5, and only the multiple of the expansion of the degree of high-quality development is recorded.

**Table 5.** Identification and analysis of key constraints.

| Parameter | Attribute | Evaluation Result |
|---|---|---|
| Irrigation water quota for farmland | − | 1.009 |
| Industrial water quota | − | 1.009 |
| ecological water | + | 0.997 |
| Water quota for rural residents | − | 1.001 |
| Water quota for urban residents | − | 1.002 |
| Consumption of energy | − | 1.011 |
| Electricity consumption | − | 1.004 |
| Cultivated area | + | 0.996 |
| Grain output | + | 1.001 |
| emissions of wastewater | − | 1.048 |
| COD concentration | − | 1.006 |
| Discharge of sewage | − | 1.006 |
| exhaust gas emission | − | 1.008 |
| carbon emission | − | 1.01 |
| Average annual growth rate of primary industry | + | 1.002 |
| Average annual growth rate of secondary industry | + | 1.003 |
| Average annual growth rate of tertiary industry | + | 1.007 |
| Length of expressways | + | 1.001 |
| number of medical and health personnel | + | 1.004 |
| growth rate of population | + | 1.001 |

From the results, it can be seen that the parameters that have a greater impact on the degree of high-quality development are the average irrigation water quota per mu of farmland, industrial water quota, energy consumption, wastewater discharge, exhaust gas emissions, carbon emissions, and the annual growth rate of the tertiary industry. The corresponding evaluation indicators are per capita water consumption, per capita energy consumption, wastewater emissions per 10,000 yuan of GDP, carbon emissions per capita, GDP per capita, proportion of tertiary industry, and number of medical and health personnel per 10,000 persons. Prioritizing these indicators when regulating high-quality development level.

3.2.2. Control Scheme Design

Based on the high-quality development regulation model constructed above, different parameters concerning the development model, organizational status, and policy factors of Henan Province in recent years have significantly influenced the model to obtain scenario development plans with different purposes. Through the simulation of the system, that is, after entering different parameters, the long-term dynamic changes of the system are predicted through the changing trend and dynamic development of the system. Finally, the simulation results are evaluated and compared with the adaptive utilization of water resources, and comprehensive countermeasures to adapt to the high-quality development of the yellow water-receiving area of Henan Province are proposed. The following six different scenarios are considered.

Status quo continuation type: maintain the status quo development and use the actual value to simulate future development without changing the model parameters.

Resource-bearing type: perpetuate the status quo and put economic development on the backburner. The goal is to protect resources, adhere to the red line of cultivated land while increasing the cultivated land area as much as possible, and minimize the use of water resources.

Ecological protection type: take ecological protection as the goal, put resource carrying and economic development in a secondary position, pay attention to urban greening, increase ecological water consumption, and improve sewage treatment rate.

Economic development type: with the goal of economic development, unilaterally pursue high-level development of economic factors and increase the development speed of various industries, without considering the constraints of resources and the environment.

Social happiness type: With the goal of people's happiness, put resource bearing and economic development in a secondary position, vigorously develop the education industry, improve medical security measures, and build supporting facilities for residents' bare life.

Comprehensive development type: the comprehensive adjustment of resource bearing–ecological health–economic development–social happiness, under the premise of resource bearing, moderate economic development while considering ecological health, but also to achieve social happiness and harmony.

Adjusting the value of the critical constraint can be realized by changing the corresponding parameters. The status quo continuation scheme mentioned refers to the high-quality development state of the region in the coming years. When there are no regulatory measures, using the ARMA model [30] to predict the future values of the parameters. Substituting them into the regulation model, you can obtain the data of each indicator in the status quo continuation state, and the parameter values of other schemes are determined by referring to the parameter values and policy provisions of the status quo continuation program.

The values of the key constraints under each type in 2025 are shown in Table 6.

**Table 6.** The values of key constraints under each type in 2025.

| Index | Status Quo Continuation Type | Resource-Bearing Type | Ecological Protection Type | Economic Development Type | Social Happiness Type | Comprehensive Development Type |
|---|---|---|---|---|---|---|
| A3 | 180.17 | 173.6 | 180.17 | 180.17 | 180.17 | 173.6 |
| A4 | 1.87 | 1.42 | 1.87 | 1.87 | 1.87 | 1.42 |
| B2 | 0.81 | 0.81 | 0.65 | 0.81 | 0.81 | 0.62 |
| B7 | 0.039 | 0.039 | 0.036 | 0.039 | 0.039 | 0.036 |
| C1 | 73,247 | 73,247 | 73,247 | 76,313 | 73,247 | 76,313 |
| C3 | 0.61 | 0.61 | 0.61 | 0.63 | 0.61 | 0.63 |
| D6 | 69.02 | 69.02 | 69.02 | 69.02 | 74.92 | 74.92 |

3.2.3. Simulation Results and Analysis

The parameter values under each scheme are regulated in the Vensim model according to the above different scheme designs. The values of each index can be obtained after running the model. The high-quality development degree of the region under each control scheme can be evaluated, and the control results under each scheme are shown in Figure 6.

It can be seen from Figure 6 that under the current situation continuation plan, the overall high-quality development degree of the yellow water-receiving area in Henan Province has increased from 0.56 to 0.77, which is significantly slower than that of 2011–2020. Through artificial regulation, the degree of high-quality regional development under each plan has improved compared with the continuation of the status quo. The improvement effect of the three schemes of resource protection, ecological protection, and social happiness is not apparent, and the regulation results are not ideal. Under the control of the economic plan, high-quality regional development increased from 0.56 to 0.79, which was a significant improvement. Under the comprehensive development-oriented regulation and control plan, the degree of high-quality regional development has increased from 0.56 to 0.8, and the improvement effect is noticeable. The optimal regulation

and control scheme is a comprehensive development plan. Under external control measures, the degree of high-quality regional development under each control scheme has improved. However, the effect is not very obvious, which also shows that the improvement potential of each system is limited. When it reaches a certain level, the value of each parameter is limited. Under traditional production and lifestyle, it is impossible to improve the degree of high-quality regional development, and it is necessary to open up new production and lifestyle and take the road of innovation. After adopting regulatory measures, the degree of high-quality regional development can reach up to 0.8, which is very close to the ideal level, indicating that external regulatory measures are necessary and can ensure high-quality regional development to a certain extent.

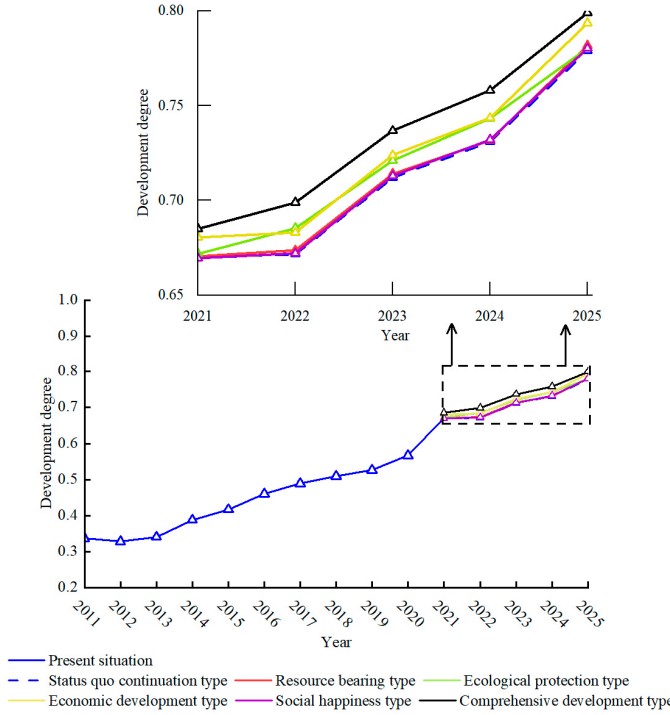

**Figure 6.** Governance result graph. Chart of regulation results under each scheme.

## 4. Conclusions

This study constructs a system-based high-quality development evaluation index system based on the research basis of the "Yellow River Strategy" by experts and scholars. It carries out quantitative research on the level of high-quality regional development. Based on the research on the mechanism of a harmonious balance between systems, a harmonious regulation model of high-quality regional development is constructed, the key constraints affecting regional development are identified by using the regulation model, and finally, the regulation model is applied to the regulation of high-quality development in the southern section of the Yellow River, and the main conclusions are as follows:

(1) From 2011 to 2020, the high-quality development degree of the Huanghuang water-receiving area in Henan Province increased from 0.34 to 0.57, and the improvement effect was pronounced; among the subsystems, the resource system was always at a high level, and the resources, ecology, and economic systems were in a steadily rising stage. The resource–ecology–economy–social system is in a non-harmonious equilibrium but is developing toward a harmonious equilibrium.

(2) Through the harmonious regulation of high-quality development, the per capita water consumption, per capita energy consumption, 10,000 yuan GDP wastewater discharge, 10,000 yuan industrial value-added exhaust gas emissions, per capita carbon

emissions, and other indicators are regulated. The high-quality regional development degree can reach 0.8, indicating that under external control measures, the degree of high-quality regional development can be effectively improved, and regional high-quality development can be realized.

(3) The results of the regulation of high-quality development of the Huanghuang water-receiving area in Henan Province show that the critical influencing factors identified by the regulation can effectively improve the degree of high-quality regional development, and the research methods are feasible and practical. However, the constructed index system involves too many indicators, and the completeness and accuracy of the primary data are high, which needs to be further studied and improved.

In this paper, a high-quality development regulation model is constructed based on the ESD model, which basically simulates the dynamic changes between systems. On one hand, due to the complex and changeable interaction relationship between systems, the analysis of the interaction mechanism between systems in this paper is simple and macroscopic, which can be carefully deepened in future research. On the other hand, the regulation model proposed in this study is an artificial regulation method based on scenario design, and dynamic regulation can be tried in the future, without setting scenarios, and the optimal high-quality development path can be obtained directly through the model.

**Author Contributions:** Conceptualization, X.Z. and C.H.; methodology, Y.Z.; software, Y.Z.; validation, X.Z., C.H., and Y.Z.; resources, C.H.; data curation, Y.Z.; writing—original draft preparation, Y.Z.; writing—review and editing, Y.Z., X.Z., and C.H. All authors have read and agreed to the published version of the manuscript.

**Funding:** This research was funded by the Major Science and Technology Projects for Public Welfare of Henan Province (No. 201300311500), Science and Technology Research Projects of Henan Province (No. 212102311156), Water Conservancy Science and Technology Research Projects of Henan Province (No. GG202042), open subject fund projects of Yellow River Sediment Laboratory of the Ministry of Water Resources (No. HHNS202005), National Natural Sciences Foundation of China (No. 52109017), The Natural Science Foundation of Henan Province (No. 222300420492), and North China University of Water Resources and Electric Power Library Postgraduate Innovation Project (No. YK-2021-42).

**Data Availability Statement:** The data presented in this study are openly available in [Henan statistical Yearbook] at [https://www.henan.gov.cn/zwgk/zfxxgk/fdzdgknr/tjxx/tjnj/].

**Conflicts of Interest:** The authors declare no conflicts of interest.

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
