# Peer review of "Research on High-Quality Development Evaluation and Regulation Model: A Case Study of the Yellow River Water Supply Area in Henan Province"

_water, doi:10.3390/w15020261_

Round 1
Reviewer 1 Report
Overall suggestion:
The authors used the single-index quantification-multi index synthesis-multi criterion integration method (SMI-P) to evaluate the high-quality development level of the southern section of the Yellow River and identified the key constraints. Using the method of embedded system dynamics, a high-quality development regulation model was established, and the changes of high-quality development level under different development scenarios were simulated. This is an interesting study, which is of great value to regional studies as a case study. It is also an essential resource for Water because achieving high-quality human and natural development is a hot and challenging current issue. It is recommended to adopt it after some revisions.
Below are some comments and suggestions for the improvement of the manuscript.
Specific suggestions:
Comment 1: The classification in Article 2.1 needs to be clarified. Please correct it.
Comment 2: The abbreviations for each system in Figure 4 and Figure 5 do not appear in the previous article, so please add them where appropriate.
Comment 3: The description in the picture must be consistent with the text. For example, the scheme name in 3.2.2 must be consistent with the legend in Figure 6.
Comment 4: Which year's regulation results are the values of key factors in Table 5? Please be specific.
Author Response
请参阅附件。

Reviewer 2 Report
Please see attached file.
